# Explicit and consistent aerosol correction for visible wavelength satellite cloud and nitrogen dioxide retrievals based on optical properties from a global aerosol analysis

Alexander Vasilkov[1], Nickolay Krotkov[2], Eun-Su Yang[1], Lok Lamsal[3], Joanna Joiner[2], Patricia Castellanos[2], Zachary Fasnacht[1], and Robert Spurr[4]

[1]Science System and Applications, Inc., Lanham, MD, USA
[2]NASA Goddard Space Flight Center, Greenbelt, MD, USA
[3]Universities Space Research Association, Columbia, MD, USA
[4]RT Solutions, Inc., Cambridge, MA, USA

**Correspondence:** A. Vasilkov (alexander.vasilkov@ssaihq.com)

**Abstract.** We discuss an explicit and consistent aerosol correction for cloud and $NO_2$ retrievals that are based on the mixed Lambertian-equivalent reflectivity (MLER) concept. We apply the approach to data from the Ozone Monitoring Instrument (OMI) for a case study over northeastern China. The cloud algorithm reports an effective cloud pressure, also known as cloud optical centroid pressure (OCP), from oxygen dimer ($O_2-O_2$) absorption at 477 nm after determining an effective cloud fraction (ECF) at 466 nm. The retrieved cloud products are then used as inputs to the standard OMI $NO_2$ algorithm. A geometry-dependent Lambertian-equivalent reflectivity (GLER), which is a proxy of surface bidirectional reflectance, is used for the ground reflectivity in our implementation of the MLER approach. The current standard OMI cloud and $NO_2$ algorithms implicitly account for aerosols by treating them as non-absorbing particulate scatters within the cloud retrieval. To explicitly account for aerosol effects, we use a model of aerosol optical properties from a global aerosol assimilation system and radiative transfer computations. This approach allows us to account for aerosols within the OMI cloud and $NO_2$ algorithms with relatively small changes. We compare the OMI cloud and $NO_2$ retrievals with implicit and explicit aerosol corrections over our study area.

## 1 Introduction

Global mapping of tropospheric trace-gas pollutants such as nitrogen dioxide ($NO_2$) and sulfur dioxide ($SO_2$) from ultraviolet (UV) and visible (Vis) spectrometers, such as the Ozone Monitoring Instrument (OMI) flying on the National Aeronautics and Space Administration (NASA) Aura satellite, has enabled many scientific studies and applications in air quality monitoring including "top-down" emissions estimates, trend studies, and assimilations into chemistry-transport models for "chemical weather" forecasts (see summary of Levelt et al., 2018). Recent progress has been facilitated by innovations in technology (i.e., satellite hyperspectral UV/Vis spectrometers with relatively high spatial resolution) as well as advances in trace-gas retrievals facilitated by development of linearized radiative transfer models (RTMs). While the trace-gas algorithms have matured greatly over the past few decades and have been scrutinized by comparisons with independent measurements from

ground- and aircraft-based platforms, there is still room for further improvement. For example, it has been long recognized that the effects of aerosols on trace-gas retrievals are significant, particularly in polluted regions, and affect both the trace gas retrieval itself as well as cloud retrievals that supply inputs to it (e.g., Martin et al., 2002; Boersma et al., 2004; Leitão et al., 2010; Castellanos et al., 2015; Lorente et al., 2017). Even for clear-sky conditions, aerosols impact trace gas retrievals in complicated ways due to different optical properties of various aerosol types and the relative vertical distributions of aerosols and gases (e.g., Castellanos et al., 2015; Chimot et al., 2016; Liu et al., 2019). While aerosol effects on cloud and trace-gas retrievals themselves have been known for some time, a globally consistent aerosol correction strategy has been hampered by two key obstacles: a lack of global distributions of aerosol optical property vertical profiles, and the need for accurate (on-line) and fast RTMs for both cloud and trace-gas retrievals that explicitly account for aerosol effects; existing RTMs tend to be computationally prohibitive in their native forms.

The retrieval of the vertical column density of a trace gas like $NO_2$ requires a detailed radiative transfer modeling that includes treatment of clouds, the surface, and aerosols. A linearized RTM is used to analytically calculate the Jacobians needed for computation of vertically resolved Air Mass Factors, $AMF(h)$, that are defined as sensitivities of satellite measured radiances with respect to a trace gas concentration at a given height $h$. While atmospheric molecular (Rayleigh) scattering limits satellite sensitivity to surface pollution, clouds and/or aerosols can either decrease (shielding effect) or enhance satellite sensitivity, depending on their optical properties and vertical distributions relative to the trace gas vertical profile (e.g., Palmer et al., 2001). Sensitivity studies suggest that weakly absorbing humidified aerosols typical of the eastern US in summer can cause $NO_2$ clear sky AMF to change by up to 8%; this is partially and implicitly accounted for in the cloud correction (Boersma et al., 2011). Lin et al. (2014, 2015) estimated much larger aerosol effects over eastern China (15-40% on annual mean $NO_2$ amounts) with large seasonal and regional variabilities.

Several studies have attempted to explicitly account for aerosol effects within limited regions. These studies have either used aerosol information from chemistry transport models (Martin et al., 2003; Lin et al., 2014, 2015), derived from the same instruments as used for the trace-gas retrievals (Chimot et al., 2019) and/or other instruments (Castellanos et al., 2015), or a combination of model and data retrieved from different instruments (Liu et al., 2019). In an analysis of the aerosol effects on $NO_2$ retrievals over South America during the biomass burning season, Castellanos et al. (2015) found 30-50% average differences in clear-sky $NO_2$ AMFs when aerosols were explicitly accounted for, but for individual pixels the AMFs could differ by more than a factor of two. Lin et al. (2014, 2015) reported better agreement with independent $NO_2$ observations over southeastern China when aerosols are accounted for using data from the GEOS-Chem model with further adjustment through MODIS monthly aerosol optical depth (AOD) dataset. Liu et al. (2019) further improved the aerosol correction for OMI tropospheric $NO_2$ retrievals over east Asia using constraints from Cloud-Aerosol Lidar with Orthogonal Polarization (CALIOP) aerosol vertical profiles. All of these studies were carried out on a regional scale owing to the high computational burden of on-line RT calculations needed to account for vertically-resolved aerosol effects within the $NO_2$ retrievals. Chimot et al. (2019) used AOD and aerosol layer height derived from the $O_2-O_2$ absorption band on the same satellite instrument (Chimot et al., 2017, 2018) as inputs with a neural network based approach to derive this information in a computationally efficient manner. Recently, Jung et al. (2019) suggested an explicit aerosol correction of the OMI formaldehyde retrievals. They

use aerosol information from the OMI UV aerosol algorithm, OMAERUV, and look-up tables of scattering weights to compute formaldehyde AMFs. Explicit aerosol effects on the cloud products are not accounted for.

Most of these studies focused on the effects of aerosol in clear sky retrievals. The effects of aerosol in the presence of overlaying cloud layers is important and Bousserez (2014) and Leitão et al. (2010) suggest that explicit account of aerosols in this case may improve $NO_2$ retrievals in such cases.

Cloud algorithms for UV/Vis sensors typically treat aerosols implicitly by providing effective (cloud + aerosol) cloud radiance fraction (CRF) and effective cloud pressure, a.k.a. cloud optical centroid pressure (OCP), both necessary inputs for calculating AMF($h$) in trace gas algorithms (e.g., Stammes et al., 2008). Thus, cloud effects on trace gas retrievals are compromised by the (unknown) aerosol effects and this may lead to errors in AMF($h$). Surface reflectivity climatologies, based on data from the same instrument, may also erroneously incorporate the effects of aerosol, for example by being too bright in order to compensate for the presence of non-absorbing aerosol. These climatologies are used as inputs by both cloud and trace-gas algorithms and therefore may produce complex errors in AMF($h$).

To explicitly account for aerosol effects on the OMI cloud and $NO_2$ retrievals, here we use three dimensional (3D) aerosol optical properties from a state-of-the-art global aerosol modeling and assimilation system and on-line RT calculations. We provide a demonstration of an envisioned global approach for a case study over a known polluted region of northeastern China. While the current approach is still computationally burdensome to apply globally, it is anticipated that faster versions of the RT code will be developed based on machine learning approaches.

In general, our approach to explicitly account for aerosol effects is similar to that used in Liu et al. (2019) and Lin et al. (2014, 2015). However, there are some significant differences. For instance, Lin et al. (2014) applied ad-hoc scaling of their global circulation model (GCM) simulation results to match local aerosol observations in order to get realistic aerosol distributions. As an alternative, we use an assimilated aerosol product (Buchard et al., 2017). One of the strengths of using the assimilated aerosol product is that it is processed on a global scale in a seamless, consistent manner. This allows for a global rather than a regional methodology as was the case in Lin et al. (2014) and Liu et al. (2019). The assimilated aerosol product provides a complete set of aerosol optical properties which include the vertically resolved aerosol layer optical depth, single scattering albedo, and phase scattering matrix computed for a given time and space location. Furthermore, the method by Lin et al. (2014) and Liu et al. (2019) is applicable to land surfaces only. We have developed a new treatment of surface BRDF for the ocean (Vasilkov et al., 2017). This approach for water surfaces has been validated in Fasnacht et al. (2019) and allows for a global and consistent processing of satellite $NO_2$ data (Lamsal et al., 2021).

The main objective of this study is to lay out and demonstrate the end to end approach of an explicit aerosol correction and apply it to a case study in a polluted region for an approach that is ultimately intended for global application. We quantify the impact of such a correction in a polluted scenario. However, we do not validate our approach with independent ground- or aircraft-based data as it is beyond the scope of this initial feasibility study.

The paper is structured as follows: Section 2 describes a general approach, assimilated aerosol parameters, surface reflectivity treatment, and the OMI cloud and $NO_2$ algorithms. Section 3 provides results and discussions of simulated aerosol effects on

NO$_2$ AMFs for modeled aerosol profiles and a case study over a polluted region of northeast Asia. Conclusions and future work are described in Section 4.

## 2  Data and Methods

### 2.1  General framework for trace-gas retrievals from satellite UV/Vis spectrometers

Figure 1 shows a conceptual framework for trace gas retrievals from a satellite spectrometer (e.g., Aura OMI); this quantifies trace gas columns by analyzing spectral features in reflected sunlight. NO$_2$ and other gases like ozone O$_3$ and SO$_2$ each have their own unique spectral absorption signature. The differential optical absorption spectroscopy (DOAS) algorithm (Platt and Stutz, 2008), converts these spectral signatures into a slant column density (SCD), the number of absorbing gas molecules along the effective photon path through the atmosphere to the satellite. The SCD is then converted into a vertical column

density (VCD), the number of gas molecules in a vertical atmospheric column, using the concept of an air mass factor (AMF) that encapsulates the relationship between the measured SCD and VCD as VCD = SCD/AMF.

Theoretically, the relationship between SCD and VCD can be defined in terms of vertically resolved Jacobians, $J(h) = -\partial ln I / \partial \tau(h)$, where $I$ is the top-of-atmosphere (TOA) radiance and $\tau(h)$ is the gaseous absorption optical thickness at altitude $h$. Generally, the AMF is calculated as

$$\text{AMF} = \int_0^\infty J(h)S(h)dh, \tag{1}$$

(Palmer et al., 2001) where $S(h)$ is the profile shape factor. For O$_2$−O$_2$, absorption is a function of the square of the pressure, and $S(h)$ is given by

$$S(h) = \sigma(h)n^2(h)/\int_0^\infty \sigma(h)n^2(h)dh, \tag{2}$$

where $\sigma(h)$ is the O$_2$−O$_2$ absorption cross-section as a function of height (because of its dependence of temperature) and $n(h)$

is the number density of O$_2$.

Figure 2 shows an overall flow of our approach. The lower part of the diagram shows the trace-gas retrieval, in our case for NO$_2$ but this could apply to other trace-gases retrieved from UV/Vis sensors. Spectral fitting is applied to both O$_2$−O$_2$ for the subsequent cloud retrieval as well as to NO$_2$. Cloud parameters are then used as inputs to the NO$_2$ VCD algorithm. The other main inputs to the VCD algorithm are the clear- and cloud-sky Jacobians. For the Jacobian calculations, surface

bidirectional reflectance distribution function (BRDF) parameters from the MODerate-resolution Imaging Spectroradiometer (MODIS) instruments are used as inputs along with the UV/Vis sensor (OMI) sun-satellite geometry as well as collocated aerosol optical properties. Details of the individual steps and input data are given below.

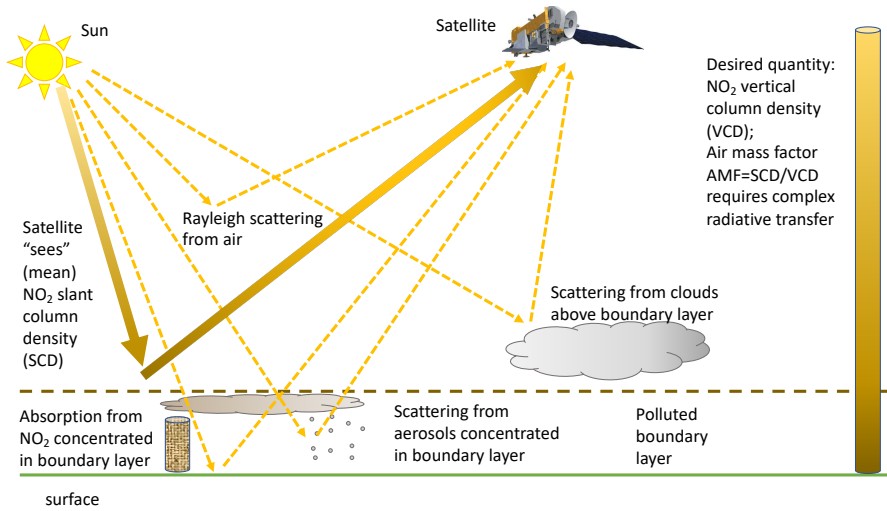

**Figure 1.** Conceptual diagram showing various paths of scattered and/or absorbed sunlight relevant to an $NO_2$ retrieval that may be observed from satellite along with standard terminology used for UV/Vis trace-gas retrievals.

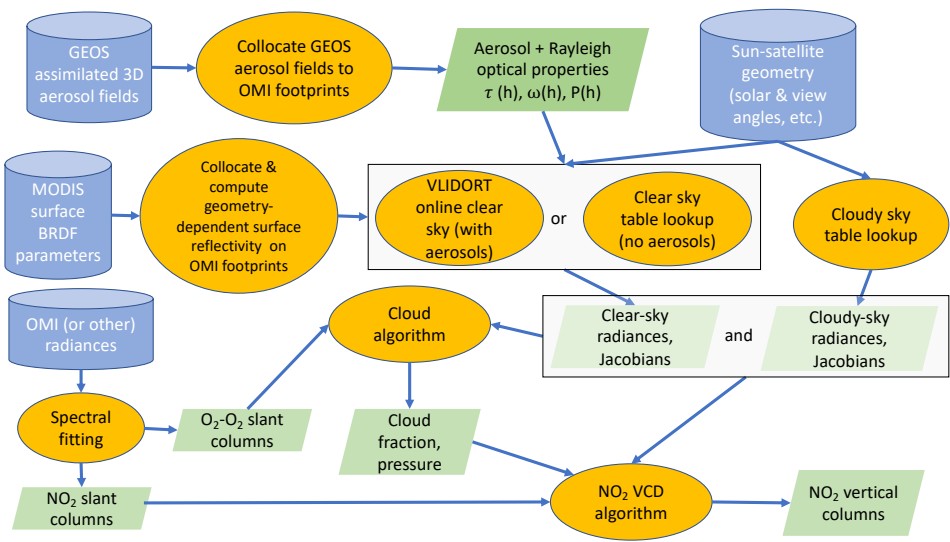

**Figure 2.** Flow diagram showing various steps and data used in our $NO_2$ retrievals.

## 2.2 Assimilated aerosol parameters

We use aerosol optical properties from the NASA Global Modeling and Assimilation Office (GMAO) Goddard Earth Observing

System version 5 (GEOS-5) system (Randles et al., 2017). The GEOS-5 global aerosol data assimilation system incorporates

information from the MODIS and recently completed a multi-decadal aerosol reanalysis, the Modern-Era Retrospective Analysis for Research and Applications version 2 (MERRA-2) (Gelaro et al., 2017), that includes assimilation of the aerosol optical depth (AOD) from various ground- and space-based remote sensing platforms (Randles et al., 2017). The analysis system is driven by a prognostic model comprising the global atmospheric circulation model, GEOS-5, radiatively coupled to the God-
125 dard Chemistry, Aerosol, Radiation, and Transport model (GOCART) (Colarco et al., 2010). The GOCART module simulates the production, loss, and transport of five types of aerosols (dust, sea salt, black carbon, organic carbon, and sulfate) treated as non-interactive external mixtures. The aerosol optical properties are described in Colarco et al. (2010) and are primarily based on the Optical Properties of Aerosols and Clouds database (Hess et al., 1998), with updates to dust properties to account for non-sphericity (Colarco et al., 2014).

The MERRA-2 global aerosol analysis data set provides vertically resolved 3D distributions of spectral aerosol layer optical depth, $\tau(h)$, single scattering albedo, $\omega_o(h)$, and scattering phase matrix, $P(h, \gamma)$ as a function of the scattering angle $\gamma$, on 72 layers from the surface to the top of the atmosphere at a native resolution of $0.5°$ latitude by $0.625°$ longitude every 3 hours. These parameters are needed for the radiative transfer (RT) computations of TOA radiance and trace gas AMFs. The MERRA-2 aerosol analysis has been evaluated against independent (not assimilated) observations from ground-, aircraft-, space-, and
ship-borne measurements (Randles et al., 2017; Buchard et al., 2017). For instance, comparisons of MERRA-2 analyzed AOD to historical (1982-1996) ship-borne measurements show that the model has a mean bias in AOD of 0.009, and a strong correlation with the observations ($r = 0.71$), while a comparison to the Marine Aerosol Network (MAN) observations from 2004-2015 showed a mean bias of 0.01 and a standard error of 0.002 ($r = 0.93$). MERRA-2 analyzed AOD was also compared to airborne High Spectral Resolution Lidar (HSRL) AOD observations during the Studies of Emissions and Atmospheric
Composition, Clouds and Climate Coupling by Regional Surveys (SEAC4RS) campaign, which consisted of several flights during August-September 2013 over North America. Compared to HSRL observations, MERRA-2 AOD has a mean bias of 0.01, and standard error of 0.005 ($r = 0.85$). The MERRA-2 aerosol analysis shows significant skill at representing dynamic global 3D aerosol distributions. For example, the MERRA-2 absorption aerosol optical depth (AAOD) and ultraviolet aerosol index (AI) compare well with OMI observations (Buchard et al., 2017).

**2.3   RT calculations**

For RT calculations here and elsewhere, we use the Vector Linearized Discrete Ordinate Radiative Transfer (VLIDORT) code (Spurr, 2006). VLIDORT computes the Stokes vector in a plane-parallel atmosphere with a Lambertian or non-Lambertian underlying surface. It has the ability to deal with attenuation of solar and line-of-sight paths in a spherical atmosphere, which is important for large solar zenith angles (SZA) and viewing zenith angles (VZA). This pseudo-spherical mode of VLIDORT
was used in all our computations including on-line calculation and generation of lookup tables. VLIDORT computes the single scattering contribution exactly in a spherically-curved atmosphere using the full scattering matrix. For multiple scattering, VLIDORT treats the direct solar beam attenuation in the pseudo-spherical approximation. This study used the delta-M scaling option to treat sharply peaked aerosol phase functions (Nakajima and Tanaka, 1988). We used 12 discrete ordinate streams in the polar hemisphere half space for the computation.

## 2.4 Surface reflectivity treatment

The Earth's surface reflectance depends on illumination and observation geometry. The surface reflection anisotropy is described by the BRDF. To account for surface BRDF in our satellite algorithms, we have introduced the concept of a surface geometry-dependent LER (GLER) in Vasilkov et al. (2017). The GLER is derived from TOA radiance computed for Rayleigh scattering and full surface BRDF for the particular geometry of a satellite instrument pixel. The TOA radiance computed by VLIDORT is then inverted to derive GLER using the following exact equation:

$$I_{TOA} = I_0 + \frac{\text{GLER} * T}{1 - \text{GLER} * S_b}, \tag{3}$$

where $I_0$ is the TOA radiance calculated for a black surface, $T$ is the total (direct+diffuse) solar irradiance reaching the surface converted to the ideal Lambertian-reflected radiance (by dividing by $\pi$) and then multiplied by the transmittance of the reflected radiation between the surface and TOA in the direction of a satellite instrument, and $S_b$ is the diffuse flux reflectivity of the atmosphere for the case of its isotropic illumination from below (Vasilkov et al., 2017). All quantities, $I_0$, $T$, and $S_b$ are calculated using a known surface pressure for a given OMI pixel. The GLER concept has been evaluated with OMI over both land (Qin et al., 2019) and ocean (Fasnacht et al., 2019).

The GLER approach provides an exact match of TOA radiances with the full BRDF approach, i.e. the TOA radiance calculated with the full surface BRDF is equal to the radiance calculated with GLER. This approach does not require any major changes to existing MLER trace gas and cloud algorithms. It simply requires replacement of the static LER climatologies with GLERs pre-computed for a specific satellite instrument. We have incorporated GLERs based on a MODIS BRDF product and use these GLERs within OMI cloud and $NO_2$ algorithms (Vasilkov et al., 2017, 2018). Climatological LER values have inevitable cloud/aerosol contamination because they are derived from TOA radiance measurements by removing the Rayleigh scattering contribution only (Kleipool et al., 2008). The cloud/aerosol contribution is minimized by selecting lower values of the residuals, however it cannot be removed completely, partially due to relatively large OMI footprint. The OMI GLER is computed using the MODIS BRDF product, which is derived from the atmospherically corrected TOA reflectance, that is after applying the MODIS cloud mask algorithm and removing aerosol scattering effects at the much higher spatial resolution of MODIS as compared with OMI. Therefore, the use of the GLER product in trace gas algorithms over heavily polluted regions greatly benefits from an explicit account of aerosols (Lin et al., 2015).

## 2.5 OMI data sets and algorithms

### 2.5.1 OMI cloud retrievals

The so-called mixed Lambert-equivalent reflectivity (MLER) concept is used in most OMI trace gas (Veefkind et al., 2006; Boersma et al., 2011; Krotkov et al., 2017) and cloud (Joiner and Vasilkov, 2006; Veefkind et al., 2016; Vasilkov et al., 2018) retrieval algorithms. It is also used in the TROPOMI $NO_2$ operational algorithm (Veefkind et al., 2012; van Geffen et al., 2019) and in the Suomi-NPP OMPS formaldehyde algorithm (González Abad et al., 2016). The MLER model treats cloud and ground as horizontally homogeneous Lambertian surfaces and mixes them using the independent pixel approximation (IPA).

According to the IPA, the measured TOA radiance is a sum of the clear sky and overcast sub-pixel radiances that are weighted with an effective cloud fraction (ECF or $f$), i.e.,

$$I_m = I_g(R_g, \text{aer})(1 - f) + I_c(R_c)\, f, \tag{4}$$

where the aerosol optical properties, $\text{aer} = [\tau(h), \omega_0(h), P(h, \gamma)]$, are from the MERRA-2 global aerosol analysis. The ECF is calculated by inverting Eq. (4) at 466 nm, a wavelength little affected by gaseous absorption or rotational-Raman scattering. The clear subpixel radiance, $I_g$, is computed on-line with the VLIDORT code for a given pixel geometry and surface pressure, $P_s$. The cloud radiance, $I_c$, is calculated using a pre-computed lookup table (LUT).

Our OMI cloud and $NO_2$ algorithms are based on the MLER model, ground and cloud being treated as Lambertian surfaces with pre-defined reflectivities. The ground reflectivity, $R_g$, is assumed to be represented by GLER that effectively accounts for surface BRDF (Vasilkov et al., 2017). The cloud reflectivity, $R_c$, is equal to 0.8 which is a common assumption (Stammes et al., 2008). Within the MLER model, here we explicitly account for aerosol for the clear-sky part of a pixel only. This is due to the simplifying treatment of cloud as an opaque surface, i.e. aerosol below the cloud does not contribute to the TOA radiance. Possible effects of aerosol above the cloud are neglected. Supporting arguments for this neglect are that aerosols are mostly observed within the planetary boundary layer, i.e. below clouds and tropospheric $NO_2$ retrievals are performed for low cloud fractions, usually for ECF<0.25.

It should be noted that a contribution of non-absorbing aerosol above a cloud with high reflectivity, as we assume within the MLER concept, to the cloud radiance is negligible. However, absorbing aerosol above the cloud can decrease the cloud radiance. Analysis of frequency of occurrence of absorbing aerosol above the cloud derived from the 12-year record (2005–2016) of OMI led to the identification of regions with frequent aerosol–cloud overlap (Jethva and Torres, 2018). Figure 5 of that work showed that the most frequent aerosol–cloud overlap occurs over the oceans where the long-range transport of aerosols plays an important role and low-level marine stratocumulus clouds are observed. This fact is also confirmed in a recent paper by Zhang et al. (2019). Those oceanic regions are of less interest for tropospheric $NO_2$ retrievals because of the small contribution of anthropogenic $NO_2$ pollution. Additionally, tropospheric $NO_2$ retrievals over the oceanic regions are sensitive to errors from other aspects of retrievals (e.g., separation of stratospheric and tropospheric components), which are more important than aerosol effects. The springtime biomass burning activities such as burning of forest, grassland and crop residue over Southeast Asia release significant amounts of smoke particles observed over the widespread cloud deck over southern China on about 20–40% of the cloudy days. $NO_2$ retrievals are typically not performed for those events owing to high cloud fractions. It is possible to flag and discard such retrievals if they were to occur in partial or thin cloud conditions using the absorbing aerosol index (Jethva and Torres, 2018). The treatment of absorbing aerosol over the cloud for $NO_2$ retrieval in such scenarios is beyond the scope of this work.

Effective cloud pressure, also called the optical centroid pressure (OCP) (Joiner et al., 2012), is derived from the $O_2-O_2$ SCD calculated using spectral fitting of the absorption band at 477 nm. The OCP, here also denoted as $P_c$, is estimated using the MLER method to compute the appropriate air mass factors (AMF) (Vasilkov et al., 2018). To solve for OCP, we invert the

following equation

$$\text{SCD} = \text{AMF}_g(P_s, R_g, \text{aer})\, \text{VCD}(P_s)\,(1-f_r) + \text{AMF}_c(P_c, R_c)\, \text{VCD}(P_c)\, f_r, \tag{5}$$

where VCD is the vertical column density of $O_2-O_2$ (VCD = SCD / AMF), $\text{AMF}_g$ and $\text{AMF}_c$ are the precomputed (at 477 nm) clear sky (subscript $g$) and overcast (cloudy, subscript $c$) subpixel AMFs, $P_s$ is the surface pressure, and $f_r$ is the cloud radiance fraction (CRF) given by $f_r = f \times I_c/I_m$. CRF is defined as the fraction of TOA radiance reflected by the cloud. In Eq. (5) the CRF is calculated at 477 nm, the center of the $O_2-O_2$ absorption band. The $O_2-O_2$ absorption cross-section depends on height because we account for its temperature dependence (Thalman and Volkamer, 2013). The clear subpixel AMF, $\text{AMF}_g$, is computed on-line with the VLIDORT code while the cloudy subpixel AMF, $\text{AMF}_c$, is calculated using a pre-computed LUT.

To solve Eq. (5) we rewrite it in the form:

$$\text{SCD}_c(P_c) \equiv \text{AMF}_c(P_c)\text{VCD}(P_c) = [\text{SCD} - \text{AMF}_g\text{VCD}_g(1-f_r)]/f_r, \tag{6}$$

where quantities on the right hand side of the equation are known, in particular, the quantity SCD is retrieved from the spectral fit of the OMI measurements around the $O_2-O_2$ absorption band at 477 nm (Vasilkov et al., 2018). Using LUT values of $\text{AMF}_c(P_c)$ and calculated $\text{VCD}(P_c)$ we then find the LUT pressure nodes $P_1$ and $P_2$ for which the following inequality is valid:

$$\text{AMF}_c(P_1)\text{VCD}(P_1) < \text{AMF}_c(P_c)\text{VCD}(P_c) < \text{AMF}_c(P_2)\text{VCD}(P_2) \tag{7}$$

or equivalently, $\text{SCD1}(P_1) < \text{SCD}_c(P_c) < \text{SCD2}(P_2)$. Then $P_c$ can be obtained by linear interpolation of P over SCD:

$$P_c = [(\text{SCD}_c - \text{SCD1})P_2 + (\text{SCD2} - \text{SCD}_c)P_1]/(\text{SCD2} - \text{SCD1}). \tag{8}$$

For a very small fraction of the ECF retrievals, ECF values can be outside the physically meaningful range of zero to one. We keep all the ECF retrievals in output orbital files thus providing the necessary diagnostic information on these physically unreasonable cases. Additionally we provide the clipped ECF retrievals, that is negative retrieved ECF values are replaced with zero and ECF values greater than one are replaced with one. Similarly, we provide these clipped CRF values as the input for the OMI NO2 algorithm. A small fraction of the cloud OCP retrievals can also appear to be unphysical (values greater than surface pressure) (Veefkind et al., 2016; Vasilkov et al., 2018). Again, we keep all OCP retrievals in output files and additionally provide clipped cloud OCP retrievals by replacing OCP values greater than the surface pressure with the actual surface pressure.

### 2.5.2 OMI NO$_2$ algorithm

The OMI NO$_2$ algorithm used here has a basis described in Krotkov et al. (2017) and references therein. Briefly, the NO$_2$ retrieval algorithm consists of determination of NO$_2$ SCD from a spectral fit of OMI-measured TOA radiance in the 402-465 nm window. The SCD is converted to VCD by using AMF calculated with various input parameters such as sun-viewing geometry, surface reflectivity, cloud pressure, cloud radiance fraction, and a priori NO$_2$ profile shapes. The characteristic

vertical distribution of $NO_2$ and separation of the AMF into tropospheric and stratospheric components allow for nearly independent estimation of the respective VCDs. The NASA OMI $NO_2$ algorithm used here utilizes a statistical approach, based on the OMI measurements, to estimate the stratospheric component (Bucsela et al., 2013).

Similar to the cloud algorithm, we explicitly account for aerosol in the calculation of tropospheric $NO_2$ clear-sky AMF only:

$$AMF_{trop} = AMF_g(P_s, R_g, aer)(1 - f_r) + AMF_c(P_c, R_c)f_r, \tag{9}$$

In Eq. (9) the CRF is calculated at 440 nm, the center of the $NO_2$ fitting window. Calculation of clear sky $AMF_g$ is carried out on-line using the VLIDORT code while calculation of cloud $AMF_c$ is performed using a LUT.

## 3 Results and Discussion

### 3.1 Simulated aerosol effects on trace-gas AMFs

Aerosols can both increase and decrease sensitivity to trace gas absorption in satellite trace gas retrievals depending on their optical properties and vertical distributions relative to the trace gas vertical profile (Lin et al., 2014; Chimot et al., 2016). Aerosol scattering and absorption may shield photons from the atmosphere below, decreasing sensitivity to trace gas absorption. This effect is particularly pronounced when the primary layer of aerosols is located above the region of atmosphere that contains the trace gas of interest. Aerosol scattering within the trace gas layer increases photon path lengths and therefore may also enhance sensitivity to trace gas absorption.

To illustrate these effects, we conduct a theoretical study of the aerosol effects on $NO_2$ scattering weights for two model aerosol profiles We perform calculations for a case where aerosols are elevated near the surface and another case where aerosols are present in an elevated layer (with a Gaussian shape and peak near 3 km altitude). For all computations, we use a single $NO_2$ profile that corresponds to a polluted region. For each aerosol profile we perform calculations for two values of $\omega_0$. We use $\omega_0 = 1.0$ for a case of non-absorbing aerosol and for the case of absorbing aerosols, we used $\omega_0 = 0.88$. For both cases we assumed that $\omega_0$ is uniform throughout the atmosphere. For these computations, we set the surface albedo to 0.05, the VZA to zero (nadir), and the SZA to $45°$. Based on the computed Jacobians, we calculate the $NO_2$ AMFs for the four different aerosol scenarios (two profiles and two values of $\omega_0$).

Figure 3(left) shows the two model aerosol profiles along with a typical vertical profile of $NO_2$ number density for polluted areas. The total aerosol optical depth (AOD) for both aerosol profiles is equal to 1.0.

Figure 3(middle) compares the Jacobians with respect to $NO_2$ layer optical depth computed for non-absorbing aerosol profiles with the Jacobian for the aerosol-free atmosphere. Here, elevated aerosol clearly exhibits enhanced sensitivity to $NO_2$ above the aerosol layer and the shielding effect below. As a result of the shielding effect of the elevated aerosol, the values of $NO_2$ AMFs are lower than that for the aerosol-free $NO_2$ AMF. The near-surface aerosol enhances the sensitivity to $NO_2$ almost for all altitudes; however, the enhanced sensitivity drops abruptly towards the surface owing to the increasing shielding effect.

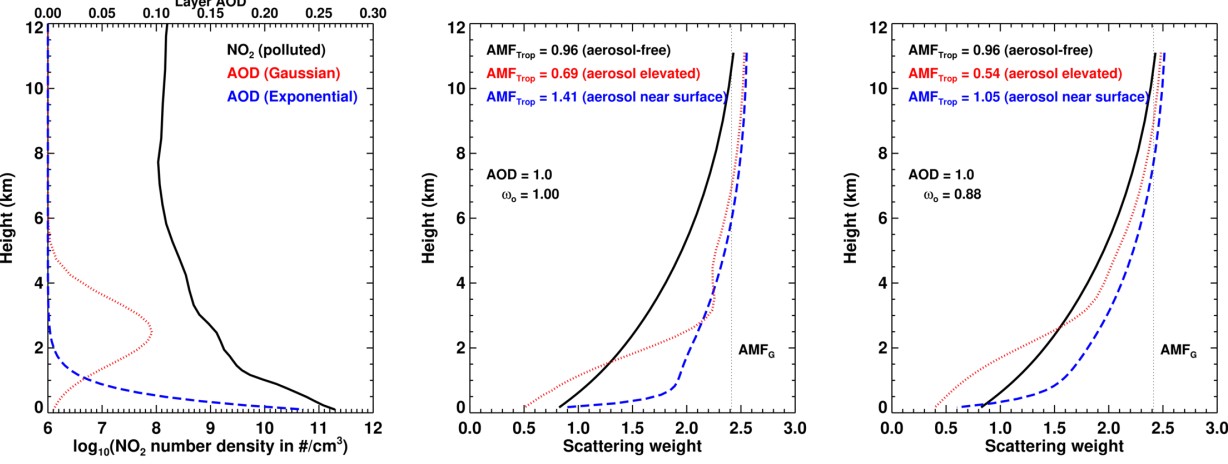

**Figure 3.** Left: Vertical profiles of tropospheric aerosols (layer aerosol optical depth (AOD), top scale) and the $NO_2$ number density (black lines, bottom scale). Middle: VLIDORT calculated $NO_2$ Jacobians for aerosol-free atmosphere (black lines) and mixed with non-absorbing (AOD=1.0, $\omega_0 = 1.0$) and absorbing (right: AOD=1.0, $\omega_0 = 0.88$) aerosols. The vertical dashed lines represent geometrical AMFs: AMF = sec(SZA) + sec(VZA), where SZA and VZA are solar and view zenith angles. Right: Similar to the middle figure but for cases of absorbing aerosols (AOD=1.0, $\omega_0 = 0.88$).

Similarly, Figure 3(right) compares the Jacobians computed for absorbing aerosols with the Jacobian for the aerosol-free atmosphere. In general, aerosol absorption decreases the $NO_2$ sensitivity for both aerosol profiles. However, the qualitative dependence of the Jacobians on height remains similar to the nonabsorbing aerosol Jacobians.

## 3.2 Case study over northeast Asia

To demonstrate our explicit aerosol correction effects on the OMI cloud and $NO_2$ retrievals, we selected a cloud-free area over land in the Shenyang region of northeastern China. Figure 4 shows a map of OMI TOA reflectance over northeastern China calculated at 440 nm for orbit 3843 on April 5, 2005. The selected cloud-free area is shown by a square on this map. The GEOS-5 MERRA-2 aerosol optical properties were collocated over nominal OMI pixels within the area. There are in total 114 OMI pixels within the selected area. The selected area has low cloud fractions (ECF<0.1), but significant aerosol loading, AOD $\approx$ 0.5-0.6 according to the MERRA-2 data set.

Figure 5 shows vertical profiles of the layer AOD, SSA, and asymmetry parameter of a scattering phase function for different OMI pixels from the MERRA-2 data set within this selected area. The asymmetry parameter characterizes the anisotropy of the phase function, i.e. a size of aerosol particles. According to the MERRA-2 aerosol analysis, most aerosol is located in the planetary boundary layer (PBL) with significant increase in aerosol loading towards the surface. There is some enhancement of aerosol loading at altitudes of about 11 km. This aerosol plume at 11 km has distinctive optical properties with increased

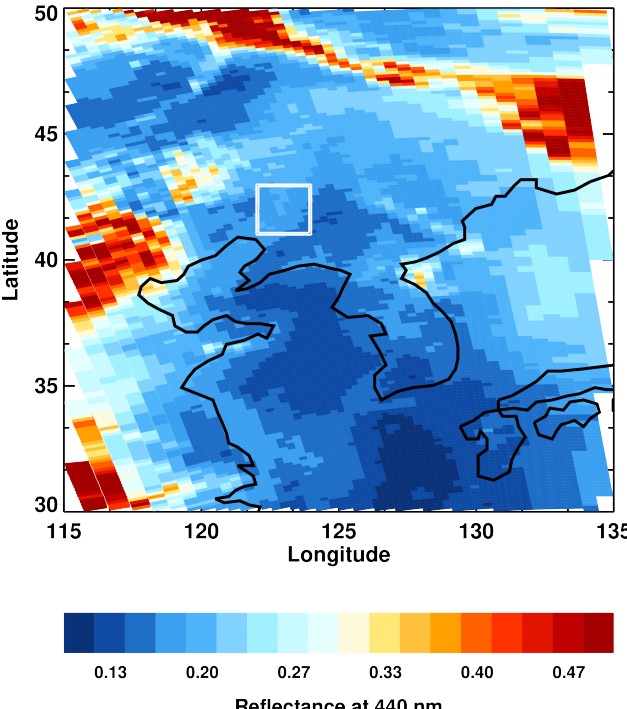

**Figure 4.** TOA reflectance at 440 nm over northeastern China for OMI orbit 3843 on 5 April 2005. The selected cloud-free region is denoted by a square.

SSA (lower aerosol absorption) and increased asymmetry parameter (larger aerosol particles). The PBL aerosol has relatively low SSA within 0.83-0.88 and slightly increased asymmetry parameter (however lower than in the high altitude plume).

NO$_2$ profiles and other model-derived information (e.g., temperature profiles, tropopause pressure) used in the computations are taken from the Global Modeling Initiative (GMI) model. The GMI simulation is driven by the meteorological fields from the MERRA-2. We use the GMI model because the simulations have been run consistently from the start of the OMI mission and this allows us to reprocess results from the entire OMI mission with the proposed aerosol correction.

Figure 6 shows both the climatological LER (Kleipool et al., 2008) and GLER for the selected area for OMI orbit 3843 on April 5, 2005. We used the climatological LER for our cloud and NO$_2$ retrievals in the following figures for the purpose of demonstrating the BRDF effects on the retrievals. It is seen from Fig. 6 that values of GLER are noticeably lower than climatological LER values because the latter represent the most probable values of LER, which implicitly account for persisting aerosol layers. On average, the difference between the climatological LER and GLER for this area is about 0.03. It should be noted that the differences include both BRDF effects and biases between the MODIS and OMI-based surface reflectance data sets. This is because the BRDF data and thus the GLERs are derived from atmospherically-corrected MODIS radiances while the climatological LERs are inherently affected by residual aerosols. Additionally, climatological LERs can be contaminated by clouds due to the substantially larger OMI pixel size as compared with MODIS footprints. Calibration differences between

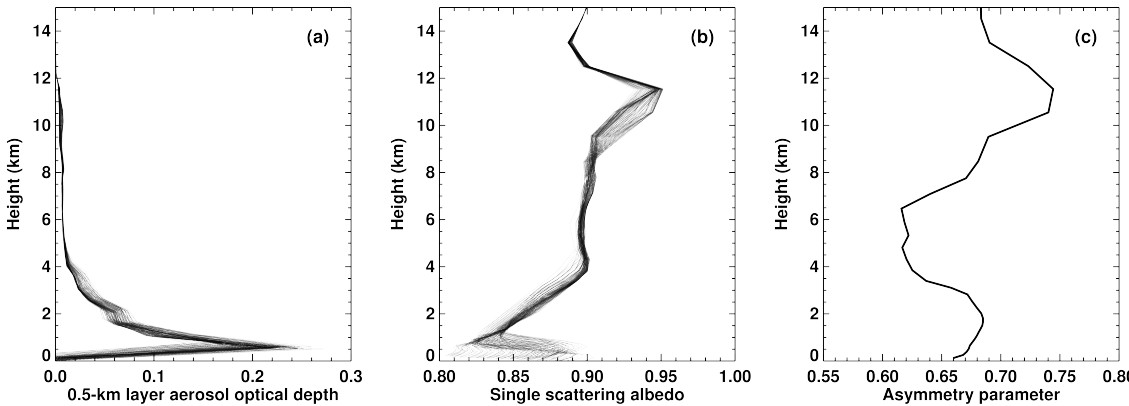

**Figure 5.** Vertical profiles of layer AOD (a), single scattering albedo (b), and asymmetry parameter (c) for different OMI pixels within the selected region.

OMI and MODIS are discussed in Qin et al. (2019) and specific details are provided in Appendix D: "Relative calibration of OMI and MODIS" of that paper. To summarize: MODIS Collection 5 radiances (used to derive BRDF kernel coefficients and thus GLER values) are higher than OMI Collection 3 radiances by approximately 1 %. A sensitivity analysis of the equation used to compute GLER shows that a 1 % error in TOA radiances will produce errors in LER of up to 0.003 in surface reflectivity. This value is much lower than the reported average difference between the climatological LER and GLER of 0.03. The atmospheric correction for MODIS band 3 used in this study has a theoretical error budget of about 0.005 reflectance units (Qin et al., 2019). Again, this error is much lower than the reported average difference suggesting that neither the calibration differences nor the MODIS atmospheric correction are major contributors to the observed difference between climatological LER and GLER.

Figure 7 compares ECF retrievals computed using climatological LERs with those computed using GLER and either implicit or explicit aerosol corrections. The comparison of ECFs retrieved with the climatological LER and the GLER and implicit aerosol correction shows the effects of replacing the surface climatological LER with the GLER only. As discussed earlier in Vasilkov et al. (2018), the GLERs are lower than the climatological LERs thus resulting in lower computed clear-sky radiances in Eq. (4) and subsequently higher retrieved ECFs. Explicit account of the aerosol contribution increases the computed clear-sky radiance thus reducing the retrieved ECF. The combined effect of GLER and explicit aerosol correction leads to ECFs slightly higher than those retrieved with the climatological LER for most pixels. The climatological LER is contaminated by aerosols and possibly clouds owing to substantially larger size of OMI pixels compared with those of MODIS data that are used for computation of GLER. That is why the lower ECFs retrieved with the climatological LER may indicate that the MERRA AOD derived for this particular day is slightly lower than climatological AOD (and possibly residual cloud optical depth) for those pixels.

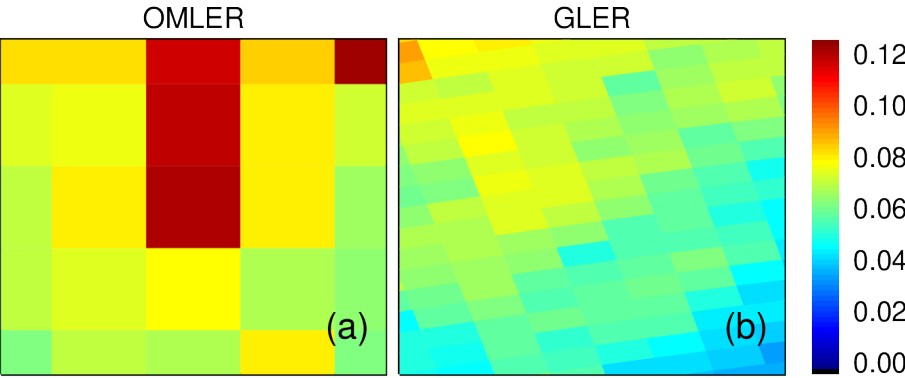

**Figure 6.** Surface LER at 440 nm over the selected area in the Shenyang region of northeastern China for OMI orbit 3843 on 5 April 2005; (a): monthly climatology at the original spatial resolution, (b): GLER computed for individual OMI pixels.

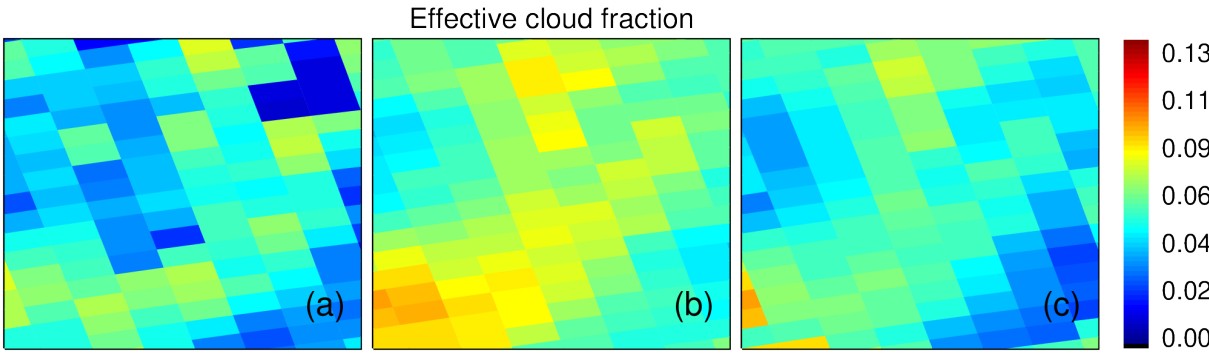

**Figure 7.** ECF retrieved with climatological surface LER (a), retrieved with GLER and implicit aerosol correction (b), and retrieved with GLER and explicit aerosol correction (c) over the selected area for OMI orbit 3843 on 5 April 2005.

Similarly, Figure 8 compares OCP retrievals computed using the climatological LER with those calculated using the GLER and either implicit or explicit aerosol corrections. The GLER effect on OCPs is mixed. For most OMI pixels, replacing the climatological LER with GLER results in lower OCPs. However for some pixels, this replacement leads to higher OCPs. It is
not straightforward to explain the GLER effect on OCP because the retrieved OCP depends on both ECF and clear-sky $O_2-O_2$ AMF, both of which are affected by replacing the climatological LER with GLER. The comparison of OCPs retrieved with either implicit or explicit aerosol correction (Fig. 8b versus Fig. 8c) shows that the explicit aerosol correction significantly increases values of the OCPs for the overwhelming majority of OMI pixels. Again, this is a complex effect with multiple factors including the ECF calculation.
Finally, Figure 9 compares tropospheric $NO_2$ VCD retrievals computed using the climatological LER with those computed using the GLER and either implicit or explicit aerosol corrections. Replacing the climatological LER with GLER significantly increases the retrieved $NO_2$ amounts as has been shown previously for polluted areas in Vasilkov et al. (2017, 2018). The

Cloud optical centroid pressure (hPa)

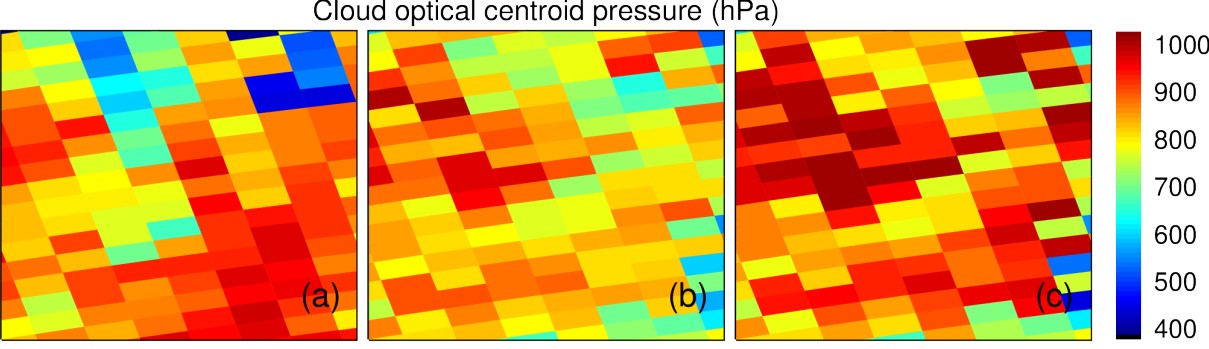

**Figure 8.** Similar to Fig. 7 but for cloud (optical centroid) pressure.

Tropospheric NO$_2$ column (molec. cm$^{-2}$)

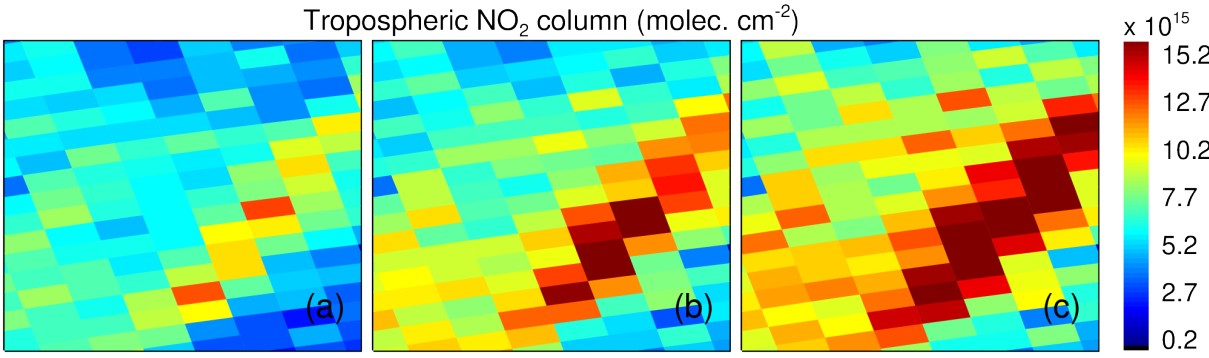

**Figure 9.** Similar to Fig. 7 but for tropospheric (trop.) NO$_2$ vertical column density.

explicit aerosol correction additionally enhances the NO$_2$ vertical column density for all OMI pixels within the selected area. This enhancement is caused by the combined effect of the explicit aerosol correction on the cloud parameters and clear-sky NO$_2$

AMFs. This aerosol correction is in line with low biases in the satellite NO$_2$ retrievals as documented in several publications (Lamsal et al., 2014; Krotkov et al., 2017; Herman et al., 2019; Choi et al., 2019). For instance, Herman et al. (2019) compared total NO$_2$ column retrievals from OMI with the ground-based Pandora at multiple sites in the US and South Korea, and found up to a factor of two lower column estimates by OMI. Assessment of OMI NO$_2$ retrievals with ground- and aircraft-based NO$_2$ observations during the DISCOVER-AQ (Deriving Information on Surface conditions from Column and Vertically Resolved

Observations Relevant to Air Quality) and KORUS-AQ (Korea-United States Air Quality Study) field campaigns suggested that OMI NO$_2$ retrievals are about 20% lower as compared to validation measurements even after accounting for the effect of a-priori NO$_2$ profiles and spatial mismatch using high-resolution NO$_2$ simulations (Choi et al., 2019). Both studies point to surface reflectivity and other factors in the NO$_2$ AMF for the low biases in OMI NO$_2$ retrievals. The application of our approach of the explicit aerosol correction to the selected area shows that the NO$_2$ increase due to the correction is in the

direction of reducing the documented low biases in the NO$_2$ retrievals with respect to ground- and aircraft-based observations.

Given that the cloud fractions are very low for the selected area (ECF < 0.1), it is reasonable to suppose that the effect of the explicit aerosol correction on the $NO_2$ enhancement is mostly caused by decreasing the clear-sky AMF. The MERRA-2 aerosol data show absorbing aerosols for the selected area (see Fig. 5) particularly for near-surface aerosol. According to our RT simulations, the absorbing aerosols mostly decrease $NO_2$ AMFs for this case. However, our preliminary analysis outside of the selected area reveals more complex picture demonstrating both shielding and enhancement aerosol effects. A global analysis of the aerosol effects will be a subject of our follow-up paper.

Figure 10 further elucidates the effect of explicit aerosol correction on cloud and $NO_2$ retrievals. It shows scatter plots of ECF, OCP, and tropospheric $NO_2$ computed with GLER and implicit versus explicit aerosol corrections. The explicit aerosol correction consistently decreases the retrieved ECF within the whole range of ECFs. This ECF decrease does not depend on an ECF value and is equal to approximately 0.015 on average. OCP changes due to the explicit aerosol correction generally depend on the value of OCP. The OCP increases with explicit account of aerosol for the overwhelming majority of pixels. This OCP increase is most pronounced for high values of OCP, i.e. for low altitude clouds. For such clouds, the OCP increases by about 100 hPa. The OCP increase is approximately 50 hPa for mid-altitude clouds with OCP of about 800 hPa. An interesting effect of the explicit aerosol correction on OCP is that OCP values for high altitude clouds are lower for a few pixels within the selected area, while in general OCP are higher for the remaining bulk of pixels. Particularly it is true for high altitude clouds with OCP values of about 500 hPa. It should be noted that an OCP error is amplified with lower cloud fraction values. This is true for all cloud pressure algorithms. In addition to OCP, we retrieve the so-called scene pressure (Vasilkov et al., 2018). In the absence of clouds and aerosols, the scene pressure should be equal to the surface pressure. A difference between the scene pressure and surface pressure can be considered as an estimate of the OCP retrieval bias. This bias is about 40 hPa. Thus an increase of 50 hPa is comparable to the expected accuracy of the OCP retrievals. However, in our work we compare the OCP retrievals with and without the explicit aerosol correction. Even though these retrievals possess bias, difference between them, e.g. increase of 50 hPa due to the implicit aerosol correction, does make sense.

The explicit aerosol correction increases the tropospheric $NO_2$ VCDs for all OMI pixels of the selected area by approximately 20% on average. This indicates that the aerosol shielding effect prevails over the effect of aerosol enhancement of photon path length for the selected area.

The uncertainties in tropospheric $NO_2$ retrievals arise from the uncertainties in $NO_2$ slant column retrievals, in the AMF calculations, and from the stratosphere-troposphere separation scheme. The uncertainty in $NO_2$ slant columns is about $0.8 \times 10^{15}$ molec cm$^{-2}$, which is typically less than 7% in high slant column cases (either over polluted areas or for observations at high solar zenith angle) and reaches up to 20% in clean areas. Uncertainties in the AMF are 20-80%, and dominate the overall retrieval uncertainties (Martin et al., 2002; Boersma et al., 2011; Bucsela et al., 2013; Lin et al., 2014) Errors in the a-priori vertical $NO_2$ profile shape, surface reflectivity, and cloud-aerosol treatment are the largest error sources (Boersma et al., 2011; Lamsal et al., 2014; Lin et al., 2014, 2015; Vasilkov et al., 2017, 2018; Liu et al., 2019). The uncertainty in the stratosphere-troposphere separation is expected to be less than $0.3 \times 10^{15}$ molec cm$^{-2}$, especially in polluted areas (Bucsela et al., 2013). Consistent with prior studies by Lin et al. (2014) and Liu et al. (2019), our study suggests that the aerosol effect over China is significant, and is similar to that of a-priori $NO_2$ profile shape and surface reflectivity.

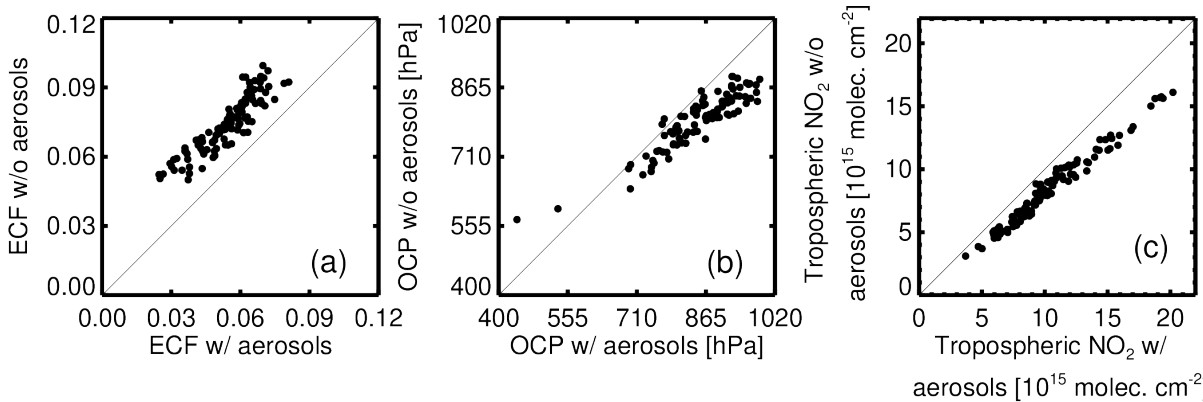

**Figure 10.** Scatter plots of retrieved quantities with implicit aerosol correction versus those retrieved with explicit aerosol correction for the selected area in OMI orbit 3843 on 5 April 2005. (a): Effective cloud fraction at 466 nm (ECF$_{466}$), (b): Cloud Optical Centroid Pressure (OCP), and (c): Tropospheric NO$_2$ vertical column density.

It should be noted that we used the vector VLIDORT code (Spurr, 2006) to calculate TOA radiances and vertically resolved O$_2$−O$_2$ and NO$_2$ Jacobians in our case study. Such calculations have been too computationally expensive for on-line use in global processing of multi-year satellite data records. A scalar approximation to the radiative transfer equation implemented using the LIDORT code is much faster than VLIDORT and save computational costs by about an order of magnitude. However the LIDORT produces errors in TOA radiance as large as 10% due to neglect of polarization effects. Recently, an artificial neural network (NN) technique to correct TOA radiances from the LIDORT to within 1% of vector-calculated radiances has been developed (Castellanos and da Silva, 2019). We plan to optimize the NN technique for the OMI cloud and NO$_2$ algorithms and extend it to calculate vertically-resolved Jacobians.

## 4 Conclusions

We discuss a new approach to explicitly account for aerosol effects on cloud and NO$_2$ retrievals. This approach can be easily incorporated into the existing operational algorithms based on the MLER concept. A main feature of the approach is that we use a complete set of aerosol optical properties which include the vertically resolved aerosol layer optical depth, single scattering albedo, and phase scattering matrix computed for a given time and space location from the global aerosol modeling and assimilation system. The surface BRDF is accounted for in the RT computations using the GLER concept (Vasilkov et al., 2017), that provides a computationally efficient method of treating BRDF in the MLER-based satellite algorithms. Comparisons of the new explicit with existing implicit aerosol correction over a polluted case study area in northeast China show that our explicit aerosol correction over polluted areas (1) decreases the retrieved ECF by 0.015 on average; (2) increases the OCP by about 100 hPa for low altitude clouds and about 50 hPa for mid-altitude clouds; and (3) increases the tropospheric NO$_2$ retrievals by about 20%. This NO$_2$ enhancement due to the explicit aerosol correction could reduce the documented

biases in the OMI $NO_2$ retrievals with respect to ground- and aircraft-based observations (Herman et al., 2019; Choi et al., 2019). It should be noted that the above estimates of the explicit aerosol correction effects on cloud and $NO_2$ retrievals are valid for the selected area. More detailed investigation of the aerosol effects on the global scale will be carried out in the future work.

Our approach requires on-line computations because it is difficult to implement a look-up table technique for inputs that include vertically-resolved optical parameters of aerosol. Currently, the on-line VLIDORT computations are not feasible for global processing of satellite data, particularly from high spatial resolution instruments such as TROPOMI and upcoming geostationary missions such as Korean Geostationary Environment Monitoring Spectrometer (GEMS), the NASA Tropospheric Emissions: Monitoring of Pollution (TEMPO), and the European Space Agency (ESA) Sentinel 4. We plan to further develop the NN technique (Castellanos and da Silva, 2019) to speed up the RT computations and apply our explicit aerosol correction to operational processing of OMI data globally.

We also plan to analyze global $NO_2$ retrievals with implicit (standard OMI $NO_2$ product) and explicit aerosol corrections and assess the impact by comparing with independent $NO_2$ observations. We plan to carry out comprehensive comparisons of our retrievals with ground- and aircraft-based NO2 observations during field campaigns such as DISCOVER-AQ and KORUS-AQ as well as with ground-based Pandora and MAX-DOAS $NO_2$ observations over various times and locations. The $NO_2$ retrievals will be performed using the measured NO2 profiles, if available, or high-resolution regional $NO_2$ simulations with implicit and explicit aerosol corrections. A reduction of the biases due to the implicit aerosol correction would prove the validity of the approach.

*Data availability.* The MODIS gap-filled BRDF Collection 5 product MCD43GF used for calculation of GLER in this paper is available at ftp://rsftp.eeos.umb.edu/data02/Gapfilled/. The OMI Level 1 data used for calculations of GLER are available at https://aura.gesdisc.eosdis.nasa.gov/data/Aura_OMI_Level1/. The OMI Level 2 Collection 3 data that include cloud, $NO_2$, and OMI pixel corner products are available at https://aura.gesdisc.eosdis.nasa.gov/data/Aura_OMI_Level2/.

*Author contributions.* AV analyzed aerosol effects on the cloud and $NO_2$ retrievals and wrote the manuscript. NK developed the GLER concept and participated in writing the manuscript. ESY performed computations of the $O_2-O_2$ and $NO_2$ scattering weights and retrievals of cloud parameters. LL applied the GLER and cloud retrievals to the $NO_2$ retrieval algorithm. JJ developed the cloud OCP concept and participated in writing the manuscript. PC calculated vertical profiles of aerosol optical properties. ZF provided collocation of GEOS-5 aerosol data onto OMI ground pixels. RS developed the VLIDORT code used for computation of the scattering weights.

*Competing interests.* The authors declare that they have no conflict of interest.

*Acknowledgements.* Funding for this work was provided by NASA through Aura core team funding as well as the Aura project and Aura Science Team and Atmospheric Composition Modeling and Analysis Program managed by Kenneth Jucks and Barry Lefer. This work was

funded in part by the $NO_2$ MEaSUREs project led by L. Lamsal, grant number 80NSSC18M0086.

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
