# Peer review of "Explicit and consistent aerosol correction for visible wavelength satellite cloud and nitrogen dioxide retrievals based on optical properties from a global aerosol analysis"

_Atmospheric Measurement Techniques, 2019_

## Referee Comment (RC1) · Anonymous Referee #1 · 24 Mar 2020

In this paper, the authors discuss the impacts of explicit aerosol corrections in OMI NO2 retrieval. They explicitly account for aerosol for the clear-sky part of a pixel by taking aerosol optical properties from the GEOS-5 global aerosol assimilation system. A case study over Shenyang, Northeastern China, is used as a specific case with high aerosol loadings to evaluate this method. This study is a part of the update in NASA NO2 product, which should be more comprehensive. I strongly encourage the authors to expand their discussion and provide more robust and convincing arguments in support of their approach. I recommend publication of this manuscript only after major revision

addressing the following comments.

General comments:

1, The explicit aerosol corrections are only applied to the clear-sky part of a pixel, which causes inconsistency between the cloud and NO2 retrieval. The studies of Lin et al., (2014) and Liu et al., (2019) have already shown the impacts of aerosols on cloud retrievals. Jethva et al., (2016) also showed absorbing aerosols were observed over clouds in spring and winter in Eastern China. So at least, the authors should discuss the impacts of explicit aerosol treatments on cloudy-sky retrievals.

2, The authors have listed plenty of comparisons in the introduction to illustrate the difference between satellite retrievals and other measurements. However, only one case study without any comparisons with ground- or aircraft-based data are discussed in the paper. The reviewer is doubt about the applicability of this method. The authors should collect some measurements and make further comparisons since it seems that the method can be applied to anywhere globally. At least, more cases should be collected to reach comprehensive/valid analysis.

Specific comments: 1, Line 3: "over norththeast Asia" should be "over northeastern Asia". Based on the context, the authors only discussed a specific case over northeastern China.

2, Line 17: "top down" should be "top-down". "assimilation" should be "assimilations"

3, Line 26: "optical properties" would be more accurate than "scattering properties"

4, Line 27: Please cite (Castellanos et al., 2014; Liu et al., 2019)

5, Line 31: What do you mean by "detailed"?

6, Line 33: I do not think the definition of the Jacobians (AK) is the same as AMF.

7, Line 47: "southeast Asia" should be "Eastern China"

[Figure]

8, Line 48: "using data from the GEOS-Chem model with further adjustment through MODIS monthly AOD dataset."

9, Line 51: Not exactly. Lin et al., (2014, 2015) and Liu et al., (2019) claimed that they used parallel RTM to ensure the efficiency of the calculation.

10, Line 175-180: See the general comment 1. Please add some specific case to help valid the argument here. "part of pixel only" should be "part of a pixel only"

11, Line 244: What causes such an enhancement? It happens in this specific case or it related to the way that GEOS-5 uses to separate the troposphere and stratosphere?

12, Based on my understanding, the procedure for deriving GLER does not include aerosol optical properties, either. GLER is an important concept in this paper. Please add the definition or give a brief introduction to it.

---

## Referee Comment (RC2) · Anonymous Referee #3 · 2 Jun 2020

This manuscript describes at method to account for aerosol effects in retrievals of tropospheric NO2. The method is based on a combination of modeled aerosol fields and measured reflectance spectra. A single case study is presented to demonstrate the method.

Main comments

The method that is presented for the aerosol correction seems highly similar to the method presented by Lin et al (2014). Therefore, the claim that is made in the conclu-

sions that this is new approach is not correct. It is a minor step forward compared to previously published work.

Although the authors claim that the method can be applied globally, there are computational problems to be resolved (line 312-319). The current description of the method is therefore incomplete for its global purpose. I believe it would be better to postpone publicatios until these problems are solved and a complete description can be given.

The one case that is presented is far too limited. Given that the authors claim to present a globally applicable method, global results for representative time periods need to be presented (e.g a few months). It is impossible to base any conclusions on the one case study that is presented.

As described in the literature and section 3.1, the aerosol effect depends on both the aerosol vertical profile and the NO2 vertical profile. Whereas it is clear that the work uses Merra-2 profiles for the aerosols, it is not clear where the NO2 profiles are coming from. This should be described clearly, and in case there are not coming from Merra -2, it should be made clear why not.

The fact that this method brings even more model information into the satellite retrievals is a concern. How can a user judge how much of the final retrieval product is model and how much is based on the measurements? This should be addressed in detail. Related to this, current NO2 retrievals include averaging kernel information which allow users to replace the assumed NO2 profile with their own profiles. It is preferable if a similar approach is implemented for the assumed aerosol profiles. This should also be addressed in the manuscript.

The sophistication of the aerosol and the cloud model seems to be out of balance. Whereas for the aerosol model a state-of-the-art aerosol model is applied, clouds are represented by a simple Lambertian clouds. The choice of this cloud model should be substantiated. Note that the retrieval results be as good as the weakest link in the chain.

Detailed comments

Section 2.5.1 This section should also describe cases for which the a-priori information is inconsistent with the measurements. For example, the following cases may occur: -The ECF becomes less than zero; -The ECF becomes larger than one; -The ECF is zero, but the SCD for O2-O2 is not consistent with aerosol profile. These are important details and all known geophysical situations should be covered in an algorithm paper.

Section 2.5.1, line 183. Details shall be provided on how the equation is solved. What numerical methods are used?

Section 2.2, line 121 How does Merra-2 deal with the very strongly non-linear growth of aerosol particles for relative humidities > 90%. This may be a frequently occurring at the top of the boundary layer for partly cloudy conditions and have significant effectson the AMF.

Section 2.3 Provide detailed information on the setup of the RT calculations wrt the aerosol optical properties, such as the number of streams used in calculations, etc.

Section 3.2, line 249 The demonstrated effect is clearly not only because of BRDF effects. The largest effect is due to different source of the surface reflectivity data. If non BRDF effects were taken into account a similar effect could be expected.

Section 3.2 line 261 Also calibration differences between OMI and MODIS and atmospheric correction in MODIS should be discussed here.

Section 3.2 line 298: "An interesting feature of the explicit aerosol correction on OCP is that the OCP can be reduced for a small fraction of the pixels." This sentence is not understood.

Section 3.2 line 297: At low cloud fractions the errors in the OCP will explode. What is the assumed error in the OCPs? How does an increase of 50 hPa relate to the expected accuracy of the OCP?

---

## Author Comment (AC2) · 2 Oct 2020

Response to reviewer #3

We thank the reviewer for the evaluation of our paper and useful comments that helped improve the manuscript. We appreciate the reviewer's time and effort in reviewing the manuscript. Below are responses to each comment. All reviewer's comments are in the standard font while the responses are in the italic font.

On behalf of the authors, Alexander Vasilkov

Main comments

The method that is presented for the aerosol correction seems highly similar to the method presented by Lin et al (2014). Therefore, the claim that is made in the conclusions that this is new approach is not correct. It is a minor step forward compared to previously published work.

*We agree that the method is similar to that presented by Lin et al. (2014) and we state this in Introduction Lines 71-72. However, there are two significant differences. First, it was necessary for Lin et al. (2014) to perform a lot of ad-hoc scaling of their GCM simulation results to match local aerosol observations in order to get realistic aerosol distributions. On the other hand, we are using a global assimilated aerosol product. One of the strengths of using the assimilated aerosol product is that this is generated by the assimilation system on a global scale in a seamless, consistent manner. One other important thing to note is that the GEOS aerosol assimilation product is constrained by MODIS and AERONET AOD observations at 550 nm. This is what differentiates our paper from Lin et al. (2014), and it is this consideration that allows for a global rather than regional methodology. Second, the method by Lin et al. (2014) is applicable to land surfaces only. We have developed a new treatment of surface BRDF for the ocean (Vasilkov et al., 2017). This approach for water surfaces is based on the GLER concept and has been validated in Fasnacht et al. (2019) and allows for a global processing of satellite instrument data.*

*We have rewritten Lines 73-75 and added the following text in Introduction:*
*"However, there are some significant differences. For instance, Lin et al. (2014) applied ad-hoc scaling of their global circulation model (GCM) simulation results to match local aerosol observations in order to get realistic aerosol distributions. On the other hand, we use an assimilated aerosol product (Buchard et al., 2017). One of the strengths of using the assimilated aerosol product is that it is processed on a global scale in a seamless, consistent manner. This allows for a global rather than a regional methodology as was the case in Lin et al. (2014) and Liu et al. (2020). The assimilated aerosol product provides a complete set of aerosol optical properties which include the vertically resolved aerosol layer optical depth, single scattering albedo, and phase scattering matrix computed for a given time and space location. Furthermore, the method by Lin et al. (2014) and Liu et al. (2020) is applicable to land surfaces only. We have developed a new treatment of surface BRDF for the ocean (Vasilkov et al., 2017). This approach for water surfaces has been validated in Fasnacht et al. (2019) and allows for a global and consistent processing of satellite $NO_2$ data."*

Although the authors claim that the method can be applied globally, there are computational problems to be resolved (line 312-319). The current description of the method is therefore incomplete for its global purpose. I believe it would be better to postpone publicatios until these problems are solved and a complete description can be given.

The one case that is presented is far too limited. Given that the authors claim to present a globally applicable method, global results for representative time periods need to be presented (e.g a few months). It is impossible to base any conclusions on the one case study that is presented.

*The main objective of this study is to lay out and demonstrate the end to end approach of an explicit aerosol correction for a case study in a polluted region for an approach that is ultimately intended for global application. However, we do not initially intend to demonstrate the aerosol correction applicability on the global scale, as it is beyond the scope of this initial feasibility study. We intended to analyze global NO₂ retrievals in the second part of this study. Based on reviewer's suggestion, we processed OMI cloud and NO₂ data globally for the same day of April 5, 2005 as in the manuscript. It appears that the aerosol effect on spatial distribution of NO₂ retrievals is even more complex than expected from the previous model study and existing literature. It is well known that the main aerosol effect on NO₂ retrievals depends on relative vertical profiles of NO₂ and aerosol as well as aerosol optical properties. For clear skies, the aerosol can both increase and decrease sensitivity of satellite instrument measurements to tropospheric NO₂. Of course, the magnitude of this effect depends on aerosol optical depth (AOD) and single scattering albedo and to lesser extent on the phase scattering matrix. For partly cloudy scenes, the presence of aerosol affects both the cloud radiance fraction (CRF) and cloud pressure, a.k.a. cloud optical centroid pressure (OCP). CRF mostly decreases but OCP can both decrease and increase and its retrieval is impacted by the derived CRF. Processing OMI data on an orbital basis reveals additional and complex features of the aerosol effects ultimately on NO₂. We think that the full explanations of the aerosol effects on cloud and NO₂ retrievals could be a topic of a second part of this study.*

*Accounting for all those considerations we have reworded Lines 284-286 as follows:*

*"The application of our approach of the explicit aerosol correction to the selected area shows that the NO₂ increase due to the correction is in the right direction of reducing the documented low biases in the NO₂ retrievals with respect to ground- and aircraft-based observations."*

*and added to the conclusions the following text:*

*"It should be noted that the above estimates of the explicit aerosol correction effects on cloud and NO₂ retrievals are valid for the selected area. More detailed investigation of the aerosol effects on the global scale will be carried out in the future work".*

As described in the literature and section 3.1, the aerosol effect depends on both the aerosol vertical profile and the NO2 vertical profile. Whereas it is clear that the work uses Merra-2 profiles for the aerosols, it is not clear where the NO2 profiles are coming from. This should be

described clearly, and in case there are not coming from Merra -2, it should be made clear why not.

*NO₂ profiles and other model-derived information used in the computations are taken from the Global Modeling Initiative (GMI) model. The GMI simulation is driven by the meteorological fields from the MERRA-2. This is clarified in the revised manuscript as follows:*

*"NO₂ profiles and other model-derived information (e.g., temperature profiles, tropopause pressure) used in the computations are taken from the Global Modeling Initiative (GMI) model. The GMI simulation is driven by the meteorological fields from the MERRA-2. We use the GMI model because the simulations have been run consistently from the start of the OMI mission and this allows us to reprocess results from the entire OMI mission with the proposed aerosol correction."*

The fact that this method brings even more model in formation into the satellite retrievals is a concern. How can a user judge how much of the final retrieval product is model and how much is based on the measurements? This should be addressed in detail. Related to this, current NO2 retrievals include averaging kernel information which allow users to replace the assumed NO2 profile with their own profiles. It is preferable if a similar approach is implemented for the assumed aerosol profiles. This should also be addressed in the manuscript.

*We do not agree with the reviewer's characterization of our approach. MERRA-2 includes assimilation of aerosol optical depth from various ground and space-based remote sensing platforms. Our motivation for using aerosol information from MERRA-2 instead of a chemical transport model (e.g., GEOS-Chem) was to include observationally constrained data which takes advantage of the relative strengths of a model and observations.*

*The NASA NO₂ standard product does not include averaging kernel, but rather it includes scattering weight (SW) profiles and tropospheric and stratospheric AMFs that allow users to re-calculate the averaging kernels and VCDs using their own a-priori NO₂ profiles. This is possible because SWs are independent of a-priori NO₂ profiles. The new SWs accounting for aerosol profiles and surface BRDF will be provided in the new NO₂ product. The users can use the new SWs to correct for their custome NO₂ profiles. However, using a different aerosol profile and surface BRDF would require re-calculating SWs. It is an inherent issue with any explicit aerosol correctio approach. We believe that our approach represents a reasonable compromise, which is feasible for global processing.*

The sophistication of the aerosol and the cloud model seems to be out of balance. Whereas for the aerosol model a state-of-the-art aerosol model is applied, clouds are represented by a simple Lambertian clouds. The choice of this cloud model should be substantiated. Note that the retrieval results be as good as the weakest link in the chain.

*For trace-gas retrievals it is important to estimate photon path lengths in the atmosphere that determine trace-gas absorption and thus affect the measured TOA radiances. Photon path lengths in a cloudy atmosphere are determined by the following most important cloud*

*parameters: the geometrical cloud fraction, the cloud optical depth, and the cloud vertical extent. Because of limited cloud informational content in TOA radiances, these three parameters cannot be retrieved simultaneously from the radiance measurements. That is why it is necessary to take on additional cloud assumptions. For instance, if a model of the Mie scattering cloud layer is used (Loyola et al., AMT, 2018), there is a need to assume a priori values for the cloud microphysical parameters and cloud vertical extent, to assume a homogeneous cloud layer and to add information about the cloud fraction from other measurements. We use a simpler model, the so-called mixed Lambertian-equivalent reflectivity (MLER) model that combines the independent pixel approximation and the treatment of cloud and ground as horizontally homogeneous, opaque Lambertian surfaces (Koelemeijer et al., JGR, 2001). The MLER model compensates for photon transport within a cloud by placing the equivalent Lambertian surface somewhere in the middle of the cloud instead of at the top (Stammes et al., JGR, 2008; Vasilkov et al., JGR, 2008; Sneep et al., JGR, 2008). As clouds are vertically inhomogeneous, the pressure of this surface does not necessarily correspond to the geometrical center of the cloud, but rather to the so called optical centroid pressure (OCP) (Joiner et al., AMT, 2012). The cloud OCP can be thought of and modeled as a reflectance-averaged pressure level reached by backscattered photons. The cloud OCP is the appropriate quantity for use in trace-gas retrievals from satellite instruments. Cloud-top pressures, e.g. those derived from thermal infrared measurements, are not equivalent to OCPs and do not provide good estimates of the required solar photon path lengths through clouds that are needed for trace-gas retrievals from UV–vis backscatter measurements (Vasilkov et al., JGR, 2008; Joiner et al., AMT, 2012). It has been demonstrated that the MLER model works reasonably well for trace-gas and cloud algorithms (Koelemeijer et al., JGR, 2001; Veefkind et al., IEEE T. Geosci. Remote, 2006; Stammes et al., JGR, 2008; Boersma et al., AMT, 2011; Bucsela et al., AMT, 2013; Veefkind et al., AMT, 2016).*

Detailed comments

Section 2.5.1 This section should also describe cases for which the a-priori information is inconsistent with the measurements. For example, the following cases may occur: -The ECF becomes less than zero; -The ECF becomes larger than one; -The ECF is zero, but the SCD for $O_2$-$O_2$ is not consistent with aerosol profile. These are important details and all known geophysical situations should be covered in an algorithm paper.

*We added the following text at the end of Section 2.5.1:*

*For a very small fraction of the ECF retrievals, ECF values can be outside the physically meaningful range of zero to one. We keep all the ECF retrievals in output orbital files thus providing the necessary diagnostic information on these physically unreasonable cases. Additionally we provide the clipped ECF retrievals, that is negative retrieved ECF values are replaced with zero and ECF values greater than one are replaced with one. Similarly, we provide these clipped CRF values as the input for the OMI $NO_2$ algorithm. A small fraction of the cloud OCP retrievals can also appear to be unphysical (values greater than surface pressure) (Veefkind et al., 2016, Vasilkov et al., 2018). Again, we keep all OCP retrievals in output files and additionally provide clipped cloud OCP retrievals by replacing OCP values greater than the surface pressure with the actual surface pressure.*

Section 2.5.1, line 183. Details shall be provided on how the equation is solved. What numerical methods are used?

*We added the following at the end of Section 2.5.1:*

*To solve Eq. (4) we rewrite it in the form:*
$SCD_c(P_c) \equiv AMF_c(P_c)*VCD(P_c) = [SCD – AMF_g*VCD_g*(1–f_r)] / f_r$
*where quantities on the right hand side of the equation are known, in particular, the quantity SCD is retrieved from the spectral fit of the OMI measurements around the $O_2$-$O_2$ absorption band at 477 nm (Vasilkov et al., 2018). Using LUT values of $AMF_c(P_c)$ and calculated $VCD(P_c)$ we then find the LUT pressure nodes $P_1$ and $P_2$ for which the following inequality is valid:*
$AMF_c(P_1)*VCD(P_1) < AMF_c(P_c)*VCD(P_c) < AMF_c(P_2)*VCD(P_2)$
*or equivalently,*
$SCD_1 (P_1) < SCD_c (P_c) < SCD_2 (P_2)$.
*Then $P_c$ can be obtained by linear interpolation of P over SCD:*
$P_c = [(SCD_c – SCD_1)*P_2 + (SCD_2 – SCD_c)*P_1] / (SCD_2 – SCD_1)$.

Section 2.2, line 121 How does Merra-2 deal with the very strongly non-linear growth of aerosol particles for relative humidities > 90%. This may be a frequently occurring at the top of the boundary layer for partly cloudy conditions and have significant effects on the AMF.

*We agree that non-linear growth of aerosol particles can be important. However this topic is beyond the scope of our paper. Here we note that MERRA-2 does account for particle hygroscopic growth. Aerosol hygroscopic growth depends on simulated relative humidity and is considered in computations of particle fall velocity, deposition velocity, and optical parameters (Randles et al., 2017).*

Section 2.3 Provide detailed information on the setup of the RT calculations wrt the aerosol optical properties, such as the number of streams used in calculations, etc.

*In Section 2.3 we have added the following:*
*VLIDORT computes the single scattering contribution exactly in a spherically-curved atmosphere using the full scattering matrix. For multiple scattering, VLIDORT treats the direct solar beam attenuation in the pseudo-spherical approximation. This study used the delta-M scaling option to treat sharply peaked aerosol phase functions (Nakajima and Tanaka, 1988). We used 12 discrete ordinate streams in the polar hemisphere half space for the computation.*

*Nakajima, T., and M. Tanaka, Algorithms for radiative intensity calculations in moderately thick atmospheres using a truncation approximation. J. Quant. Spectrosc. Radiat. Transfer, 40, 51-69, 1988.*

Section 3.2, line 249 The demonstrated effect is clearly not only because of BRDF effects. The largest effect is due to different source of the surface reflectivity data. If non BRDF effects were taken into account a similar effect could be expected.

*This is true and it was discussed in Vasilkov et al. (2017). Here we have added the following:*

*"It should be noted that the differences include both BRDF effects and biases between the MODIS and OMI-based surface reflectance data sets. This is because the BRDF data and thus the GLERs are derived from atmospherically-corrected MODIS radiances while the climatological LERs are inherently affected by residual aerosols. Additionally, climatological LERs can be contaminated by clouds due to the substantially larger OMI pixel size as compared with MODIS footprints."*

Section 3.2 line 261 Also calibration differences between OMI and MODIS and atmospheric correction in MODIS should be discussed here.

*We added the following:*

*"Calibration differences between OMI and MODIS are discussed in Qin et al. (2019) and specific details are provided in Appendix D: "Relative calibration of OMI and MODIS" of that paper. To summarize: MODIS Collection 5 radiances (used to derive BRDF kernel coefficients and thus GLER values) are higher than OMI Collection 3 radiances by approximately 1%. A sensitivity analysis of the equation used to compute GLER shows that a 1% error in TOA radiances will produce errors in LER of up to 0.003 in surface reflectivity. This value is much lower that the reported average difference between the climatological LER and GLER of 0.03. The atmospheric correction for MODIS band 3 used in this study has a theoretical error budget of about 0.005 reflectance units (Qin et al., 2019). Again, this error is much lower than the reported average difference suggesting that neither the calibration differences nor the MODIS atmospheric correction are major contributors to the observed difference between climatological LER and GLER."*

Section 3.2 line 298: "An interesting feature of the explicit aerosol correction on OCP is that the OCP can be reduced for a small fraction of the pixels." This sentence is not understood.

*We have reworded this sentence as follows: "An interesting effect of the explicit aerosol correction on OCP is that OCP values for high altitude clouds are lower for a few pixels within the selected area, while in general OCP are higher for the remaining bulk of pixels."*

Section 3.2 line 297: At low cloud fractions the errors in the OCP will explode. What is the assumed error in the OCPs? How does an increase of 50 hPa relate to the expected accuracy of the OCP?

*That is correct. To clarify the issue we have added the following text:*

*"An OCP error is amplified with lower cloud fraction values. This is true all cloud pressure algorithms. In addition to OCP, we retrieve the so-called scene pressure (Vasilkov et al., 2018). In the absence of clouds and aerosols, the scene pressure should be equal to the surface pressure. A difference between the scene pressure and surface pressure can be considered as estimates of the OCP retrieval bias. This bias is about 40 hPa. Thus an increase of 50 hPa is comparable to the expected accuracy of the OCP retrievals. However, in our work we compare the OCP retrievals with and without the explicit aerosol correction. Even though these retrievals*

*possess bias, difference between them, e.g. increase of 50 hPa due to the implicit aerosol correction, does make sense."*

---

## Author Comment (AC1)

Response to reviewer #1

We thank the reviewer for his/her evaluation of our paper and useful comments that helped improve the manuscript. We appreciate reviewer's time and effort in reviewing the manuscript. Below are responses to each comment. All reviewer's comments are in the standard font while the responses are in the italic font.

On behalf of the authors, Alexander Vasilkov

General comments:

1, The explicit aerosol corrections are only applied to the clear-sky part of a pixel, which causes inconsistency between the cloud and NO2 retrieval.

*We do not think that there is inconsistency between the cloud and NO$_2$ retrieval. Both cloud and NO$_2$ algorithms make use of the MLER model based on the independent pixel approximation with the same treatment of surface BRDF and aerosol in the clear-sky part of a pixel. The cloudy-sky part of a pixel is treated in the same manner as an opaque Lambertian surface with the same reflectivity of a commonly-accepted value of 80%. The basic assumption here is that aerosol is affecting the clear-sky part of the pixel only and is negligible in the cloudy part of the pixel. This assumption would fail for absorbing aerosol present above a cloud layer as discussed in the response to the next comment.*

The studies of Lin et al., (2014) and Liu et al., (2019) have already shown the impacts of aerosols on cloud retrievals. Jethva et al., (2016) also showed absorbing aerosols were observed over clouds in spring and winter in Eastern China. So at least, the authors should discuss the impacts of explicit aerosol treatments on cloudy-sky retrievals.

*It should be noted that NO$_2$ retrievals for cloudy sky conditions are highly uncertain and uncertainty can reach up to 100% (Boersma et al., 2011, Bucsela et al., 2013). Therefore, we recommend using NO$_2$ retrievals for clear and partially cloudy conditions only. Our focus here is on retrievals for low effective cloud fractions only, typically less than 0.25. We think that non-absorbing aerosol above the cloud with high reflectivity can be neglected. However, we agree that the impact of absorbing aerosol above the cloud is important and should be discussed.*

*We added the following in Sect. 2.5.1:*
*"It should be noted that a contribution of non-absorbing aerosol above a cloud with high reflectivity, as we assume within the MLER concept, to the cloud radiance is negligible. However, absorbing aerosol above the cloud can affect the cloud radiance. Analysis of frequency of occurrence of absorbing aerosol above the cloud derived from the 12-year record (2005–2016) of OMI led to the identification of regions with frequent aerosol–cloud overlap (Jethva et al., 2018). Figure 5 of that work showed that the most frequent aerosol–cloud overlap occurs over the oceans where the long-range transport of aerosols plays an important role and low-level marine stratocumulus clouds are observed. This fact is also confirmed in a recent paper by Zhang et al. (2019). Those oceanic regions are of less interest for tropospheric NO2*

*retrievals because of the small contribution of anthropogenic $NO_2$ pollution. Additionally, tropospheric $NO_2$ retrievals over the oceanic regions are prone to errors from other aspects of retrievals (e.g., separation of stratospheric and tropospheric components), which are more important than aerosol effects. The springtime biomass burning activities such as burning of forest, grassland and crop residue over Southeast Asia release significant amounts of smoke particles observed over the widespread cloud deck over southern China on about 20%–40% of the cloudy days. $NO_2$ retrievals are typically not performed for those events owing to high cloud fractions. It is possible to flag and discard such retrievals if they were to occur in partial or thin cloud conditions using the absorbing aerosol index (Jethva et al., 2018). The treatment of absorbing aerosol over the cloud for $NO_2$ retrieval in such scenarios is beyond the scope of this work."*

*Jethva, H., Torres, O., Ahn, C. A 12-year long global record of optical depth of absorbing aerosols above the clouds derived from the OMI/OMACA algorithm. Atmos. Meas. Tech., 11, 5837–5864, 2018.*
*Zhang, W., Deng, S., Luo, T., Wu, Y., Liu, N., Li, X., Huang, Y. and Zhu, W.: New global view of above-cloud absorbing aerosol distribution based on CALIPSO measurements, Remote Sens., 11, 2396; doi:10.3390/rs11202396, 2019.*

2, The authors have listed plenty of comparisons in the introduction to illustrate the difference between satellite retrievals and other measurements. However, only one case study without any comparisons with ground- or aircraft-based data are discussed in the paper. The reviewer is doubt about the applicability of this method.

*There is no doubt that explicit aerosol correction will reduce OMI low bias compared with ground-based data documented in numerous previous publications. Figure 9 compares current operational (b) and future (c) OMI tropospheric $NO_2$ retrievals. It shows that explicit aerosol correction enhances tropospheric $NO_2$ VCDs for all OMI pixels, which goes in right direction to mitigate general low bias in satellite $NO_2$ retrievals.*

The authors should collect some measurements and make further comparisons since it seems that the method can be applied to anywhere globally. At least, more cases should be collected to reach comprehensive/valid analysis

*Extensive OMI $NO_2$ comparisons with ground-based and aircraft measurements have been recently documented by our group (e.g. Choi et al., AMT, 2019) as well as other groups. Consensus has been reached that all current satellite tropospheric $NO_2$ measurements (OMI/GOME-2/TROPOMI) have low bias for highly polluted environments (Herman et al., AMT, 2019). Additional new comparisons are beyond the scope of this paper and are subject of a separate paper by our group (Lamsal et al., 2020, in preparation).*

*What this paper demonstrates, is that aerosol related uncertainties of current OMI (and for that matter TROPOMI) operational cloud (version 2.0) and tropospheric $NO_2$ (version 4.0) products can be unacceptably large under certain polluted conditions (e.g., Figs. 8-10), which certainly justifies consideration for implementation in the next version. Before operational*

*implementation, more rigorous comparison with a large collection of available ground-based data will be conducted.*

*Indeed, since we have already implemented GLER surface reflectance (which, by itself is explicitly aerosol corrected) in the current OMI operational algorithms, the next necessary step will be implementing explicit aerosol treatment in both cloud and $NO_2$ retrievals to make them consistent. Thus, the explicit aerosol correction is not only physically based, but indeed required for self-consistency of both cloud and $NO_2$ retrievals, which is necessary logical step prior to comparisons with ground-based and aircraft data.*

Specific comments:

1, Line 3: "over norththeast Asia" should be "over northeastern Asia". Based on the context, the authors only discussed a specific case over northeastern China.

*Corrected.*

2, Line 17: "top down" should be "top-down". "assimilation" should be "assimilations"

*Corrected.*

3, Line 26: "optical properties" would be more accurate than "scattering properties"

*Agree and done.*

4, Line 27: Please cite (Castellanos et al., 2014; Liu et al., 2019)

*Added.*

5, Line 31: What do you mean by "detailed"?

*We mean "modeling that includes treatment of clouds, the surface, and aerosols" as it is stated in the sentence.*

6, Line 33: I do not think the definition of the Jacobians (AK) is the same as AMF.

*Agree. We clarified this by replacing "or" by "needed for calculation of".*

7, Line 47: "southeast Asia" should be "Eastern China"

*Corrected.*

8, Line 48: "using data from the GEOS-Chem model with further adjustment through MODIS monthly AOD dataset."

*Added.*

9, Line 51: Not exactly. Lin et al., (2014, 2015) and Liu et al., (2019) claimed that they used parallel RTM to ensure the efficiency of the calculation.

*We mean that even with parallel RTM computations the cited studies were not carried out on a global scale as needed in operational processing of satellite instrument data. That is why we state that "these studies were carried out on a regional scale" in this sentence.*

10, Line 175-180: See the general comment 1. Please add some specific case to help valid the argument here. "part of pixel only" should be "part of a pixel only"

*We added some discussion here (see the answer to general comment #1). The missed article is added.*

11, Line 244: What causes such an enhancement? It happens in this specific case or it related to the way that GEOS-5 uses to separate the troposphere and stratosphere?

*This relatively small aerosol enhancement at altitudes of 11 km is thought to happen in this specific case. An exact cause of the enhancement is not clear.*

12, Based on my understanding, the procedure for deriving GLER does not include aerosol optical properties, either. GLER is an important concept in this paper. Please add the definition or give a brief introduction to it.

*You are right, GLER does not include aerosol contribution. It is calculated using atmospherically corrected BRDF data. We stated this in Sect. 2.4. We agree that adding the definition of GLER will be helpful. We added the definition and corresponding equation in Sect. 2.4. Lines 144-146 read now:*

*"The GLER is derived from TOA radiance computed for Rayleigh scattering and full surface BRDF for the particular geometry of a satellite instrument pixel. The TOA radiance computed by VLIDORT is then inverted to derive GLER using the following exact equation:*

$$I_{TOA} = I_0 + GLER*T / (1 - GLER*S_b),$$

*where $I_0$ is the TOA radiance calculated for a black surface, T is the total (direct+diffuse) solar irradiance reaching the surface converted to the ideal Lambertian-reflected radiance (by dividing by $\pi$) and then multiplied by the transmittance of the reflected radiation between the surface and TOA in the direction of a satellite instrument, and $S_b$ is the diffuse flux reflectivity of the atmosphere for the case of its isotropic illumination from below (Vasilkov et al., 2017). All quantities, $I_0$, T, and $S_b$ are calculated using a known surface pressure for a given OMI pixel. The GLER concept has been evaluated with OMI over both land (Qin et al., 2019) and ocean (Fasnacht et al., 2019)."*

---

## Author Response (AR2)

Response to the editor.

We thank the editor for the evaluation of our paper and useful comments that helped improve the manuscript. We appreciate editor's time and effort in reviewing the manuscript. Below are responses to each comment. All editor's comments are in the standard font while the responses are in the italic font.

We also added the following reference to a recently published paper in Line 84 of Introduction: Lamsal, L. N., Krotkov, N. A., Vasilkov, A., Marchenko, S., Qin, W., Yang, E.-S., Fasnacht, Z., Joiner, J., Choi, S., Haffner, D., Swartz, W. H., Fisher, B., and Bucsela, E.: Ozone Monitoring Instrument (OMI) Aura nitrogen dioxide standard product version 4.0 with improved surface and cloud treatments, *Atmos. Meas. Tech*., 14, 455–479, https://doi.org/10.5194/amt-14-455-2021, 2021.

On behalf of the authors, Alexander Vasilkov

The main weakness of this paper is the limited data base presented and absence of any validation, which makes it difficult to judge on the suitability/added-value of the proposed approach. Also the approach is not innovative per se (similar approaches have already been published) but it includes new elements with a high potential for larger scale application (if the performance issue is solved). Despite these elements, I appreciate the completeness of the responses to the reviewers and the efforts done to improve the manuscript. Also the justification given for a 2-step publication strategy seems acceptable to me. So I support publication after attention to the few minor comments below:

*Thank you for supporting publication of our manuscript. We agree that the approach is similar to that has already used in a few publications. However, there are two significant new elements which allows for a global rather than regional methodology. First, we are using a global assimilated aerosol product constrained by MODIS AOD observations. Second, we developed a new treatment of surface BRDF for the ocean that allows a global processing of satellite instrument data.*

pg. 3, l. 87: although validation goes beyond the scope of the paper, please indicate in which way you would proceed to validate the proposed aerosol correction

*We extended the future work paragraph (Lines 420-430) by replacing it with the following text:*

*"We also plan to analyze global $NO_2$ retrievals with implicit (standard OMI $NO_2$ product) and explicit aerosol corrections and assess the impact by comparing with independent $NO_2$ observations. We plan to carry out comprehensive comparisons of our retrievals with ground- and aircraft-based $NO_2$ observations during field campaigns such as DISCOVER-AQ and KORUS-AQ as well as with ground-based Pandora and MAX-DOAS $NO_2$ observations over various times and locations. The $NO_2$ retrievals will be performed using the measured $NO_2$ profiles, if available, or high-resolution regional $NO_2$ simulations with implicit and explicit*

*aerosol corrections. A reduction of the biases due to the implicit aerosol correction would prove the validity the approach."*

pg. 12, l. 306: note that the OMI LER does not provide the minimum LER but the 'most probable' LER, which is meant to implicitly account for the effect of persisting aerosol layers. The observed difference is therefore in a way expected.

*We agree and rewrote the corresponding sentence as follows:*
*"It is seen from Fig. 6 that values of GLER are noticeably lower than climatological LER values because the latter represent the most probable values of LER, which implicitly account for persisting aerosol layers."*

pg. 16, l. 63: contrary to what is written (and based on Fig.10), I understand that the aerosol correction decreases the retrieved ECF

*Thank you very much. You are right. We correctly stated in Conclusions that "… our explicit aerosol correction over polluted areas (1) decreases the retrieved ECF by 0.015 on average;" but mistakenly wrote "increases" in this Line. We corrected this.*

Typos:

pg. 3, l. 77: replace 'On the other hand...' by 'As an alternative...'
*Done.*
pg. 11, l. 89: there are 'in' total 114 OMI pixels...
*Added.*
pg. 13, 315: ...this value is much lower 'than' the reported average...
*Corrected.*
pg. 16, l. 371: ... is true 'for' all cloud pressure algorithms...
*Added.*
pg. 16, l. 373: ... can be considered as 'an estimate' of the OCP rerieval bias...
*Corrected.*